# Monotonic models for real-time dynamic malware detection

**Alexander Chistyakov**[1]**, Ekaterina Lobacheva**[2]*****, Alexander Shevelev**[1]**, Alexey Romanenko**[1]
[1]Detection Methods Analysis Group, Kaspersky Lab
[2]Samsung-HSE Laboratory, National Research University Higher School of Economics
Moscow, Russia
`{Alexander.Chistyakov,Alexander.Shevelev,`
`Alexey.Romanenko}@kaspersky.com, elobacheva@hse.ru`

## Abstract

In dynamic malware analysis, programs are classified as malware or benign based on their execution logs. We propose a concept of applying monotonic classification models to the analysis process, to make the trained model's predictions consistent over execution time and provably stable to the injection of any noise or 'benign-looking' activity into the program's behavior. The predictions of such models change monotonically through the log in the sense that the addition of new lines into the log may only increase the probability of the file being found malicious, which make them suitable for real-time classification on a user's machine. We evaluate monotonic neural network models based on the work by Chistyakov et al. (2017) and demonstrate that they provide stable and interpretable results.

## 1  Introduction and Motivation

Malware (i.e. malicious software) detection is an important and challenging task for the cybersecurity industry. One of the main approaches for detecting malware is a dynamic one, in which an investigated code is executed in a controlled environment and a prediction about the malware or benign label is made based on the program's execution trace. In this paper we focus on the application of machine learning methods for dynamic analysis in a real-time scenario and their stability w.r.t. code obfuscation. A real-time scenario implies that the detection task is being continuously solved while the program is running on a user's machine, and its execution is interrupted as soon as predicted probability of maliciousness becomes high. Therefore it is crucial to detect malicious behavior as early as possible to prevent or at least reduce the damage.

In the dynamic analysis the program's behavior trace is usually represented with a log of the observed system events or API calls (function name, arguments, and, optionally, a return value). There are several machine learning techniques to classify these logs as malware or benign. Most of them first extract some features based on n-grams of events, links between APIs and their arguments, or the behavior patterns in the graph representation of the log, and then apply a classifier such as neural net or boosting (Bayer et al., 2009; Berlin et al., 2015; Huang & Stokes, 2016; Salehi et al., 2017; Chistyakov et al., 2017). Some methods also exploit the sequential nature of the logs and apply recurrent neural networks (Pascanu et al., 2015; Kolosnjaji et al., 2016).

All these methods operate with full logs and do not directly aim to predict correct labels for the log's prefixes, therefore their predictions through the program's execution time may be inconsistent. This complicates the use of any of them in the real-time scenario – because at one moment of the execution the method may be sure that the program is malicious, and at the next moment the prediction may become benign. Extending the training dataset with log's prefixes cannot solve this problem, because labels for the prefixes of malicious logs are not defined (malicious program may start its main payload only after a long period of execution). Moreover, there are no specific limitations on existing methods, which guarantee that no activities in the log are used as 'benign' features. A feature is 'benign' for some model if its presence in the log brings the prediction of the

---

*Most of the work was done when Ekaterina Lobacheva worked at Kaspersky Lab.

model closer to benign. Such features should not be employed in the malware detection because they may be easily used to construct adversarial examples. For example, if starting a process from a standard directory is the 'benign' feature for the detector, then a malicious program can deceive the model by starting several processes from that directory in addition to its usual functionality.

In this paper, we propose to modify the dynamic detection techniques by making both feature extraction and classification monotonic in the sense that the addition of new lines into the log may only increase the probability of the file being found malicious. This condition results in the monotonically increasing predicted probability of maliciousness w.r.t running time, which makes the predictions consistent through the program's execution. Hence for a benign file, the prediction is benign for all moments of time and for a malware file, the prediction becomes malware at some point and remains so until the end of the log. Additionally, this condition restricts the use of 'benign' features making predictions stable w.r.t. the injection of any new functionality in the program's behavior.

In order to demonstrate that such modification is reasonable, we apply it to an end-to-end neural network model, based on the work by Chistyakov et al. (2017). However, the technique is general and can be adapted for different models both to modify a feature extraction part and a classifier, such as a neural network (Sill, 1997; Daniels & Velikova, 2010; You et al., 2017), a decision tree (Potharst & Feelders, 2002), or boosting. Our experiments show that even though the monotonic model experiences some accuracy drop in a full log classification task, it works consistently in the real-time scenario and its predictions are very interpretable, because they indicate after which events in the log the model starts to classify the program as malware.

## 2 MONOTONIC CLASSIFICATION MODEL FOR LOGS

The non-monotonic classification model for logs by Chistyakov et al. (2017) is based on a behavior graph representation of the log, in which nodes correspond to event types and arguments occurring in the log, and edges represent the occurrence of the corresponding event type and the argument in the same line of the log. To construct a feature representation of such graph authors extract behavior patterns from this graph (specific subsets of connected event types and arguments), pretrain a compact feature representations for these patterns with linear autoencoder and then aggregate features of patterns into the feature representation of a graph using dynamic pooling operations (min, max and average). As a final classifier authors use XGBoost.

As a baseline in this paper we use a slightly different version of the non-monotonic model. We replace XGBoost with a neural network and train the whole model in an end-to-end manner, so instead of pretraining pattern features with autoencoder we add an embedding layer into the model. This makes the monotonic modification of the model more straightforward and additionally accelerates the training procedure. Our experiments show that this version achieve the same results as the original one.

In the non-monotonic model only the step of behavior graph construction is monotonic because the addition of new events to a log may result only in the addition of new nodes or edges to a graph. All the other steps need modifications. To make the pattern extraction step monotonic, we impose a following constraint on pattern's definition: there is no argument outside of the pattern which is connected to all the event types from this pattern. As a result, any set of event types from the graph with all the arguments, that they share, is a pattern. Such a step is monotonic because if the new argument is added to a graph, then it is simply added to some patterns, and if the new event type is added to the graph, then the new patterns appear without removing the existing ones. To make the feature extraction steps monotonic we replace the weight matrix $W$ in the embedding layer by its element-wise absolute value $|W|$ and use just the max-pooling operation since it is the only monotonic option. Finally, to make the classifier monotonic we try some existing monotonic versions of neural networks – min-max networks (Daniels & Velikova, 2010) and lattice networks (You et al., 2017). We also implement our own version based on the same trick for weight matrices as in the embedding layer and using monotonic nonlinear functions.

Chistyakov et al. (2017) show in the experiments that their model provides higher accuracy if additional counter features are used. We also use these features because they are monotonic. We concatenate the counter features and the log features obtained after the dynamic max-pooling step into one vector, and use it in the classification step.

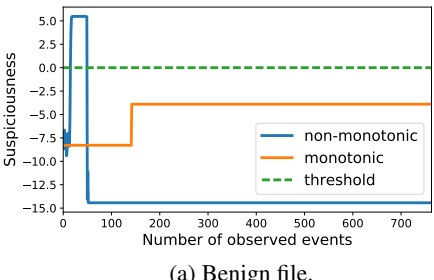
(a) Benign file.

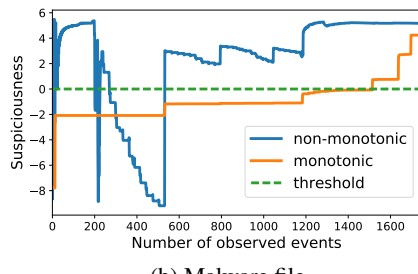
(b) Malware file.

Figure 1: Predictions of monotonic and non-monotonic models in the real-time scenario. On the vertical axis the pre-activation of the final neuron of the network is shown (the higher, the closer to the malware class). The values are shifted for each model in such way that the zero value corresponds to a classification threshold.

Table 1: AUC-ROC of monotonic and non-monotonic models in two different scenarios: the full log classification and the real-time classification.

| Scenario | Non-mon. | Mon. linear | Mon. deep | Mon. min-max |
| --- | --- | --- | --- | --- |
| Full logs (AUC-ROC) | **0.999998** | 0.987430 | 0.992089 | 0.993811 |
| Real-time (AUC-ROC) | 0.511195 | 0.987430 | 0.992089 | **0.993811** |

## 3 EXPERIMENTS

In this section we compare the baseline non-monotonic model with several variations of monotonic model. In all variations the feature extraction part is the same, but classifiers are different: we implement a linear and a deep networks with modified weight matrices and a min-max network (Daniels & Velikova, 2010). We also tried a lattice network (You et al., 2017), but its training was unstable and it showed significantly worse results than the other variations. All the models are end-to-end trainable neural networks. Baseline non-monotonic model and monotonic model with deep neural network as a classifier have the same architecture. Details about network architectures are described in Appendix A. Since there are no large publicly available datasets of the program's execution logs, 2.9M train objects and 1.2M test objects were collected for the experiments from our in-lab sandbox.

First, we compare similar non-monotonic and monotonic models qualitatively. We run both models in the real-time scenario in which the prediction is made after each new line in the log. The typical results for one malware and one benign file are shown in Figure 1. When the non-monotonic model sees the full log it makes the right prediction, but predictions for prefixes of the log are inconsistent and may change over time from malware to benign and vice versa several times. Predictions of the monotonic model grow with time monotonically, and therefore this model is much more suitable for the real-time scenario. Moreover, predictions of the monotonic model usually go up on very interpretable lines of the log, such as writing to the autorun, saving an URL corresponding to some cryptocurrency, and so on. Examples of such interpretations are presented in Appendix B.

To compare the models quantitatively, we use the AUC-ROC measure. We compare the models both in the real-time scenario and in a full log classification. In the real-time scenario, the joint prediction for the log is computed as a maximum prediction on all the prefixes of this log. For monotonic models, predictions in both scenarios are the same, because the real-time prediction reaches its maximum on the full log. For the non-monotonic model, the real-time joint prediction is always greater or equal to the full log prediction. Obtaining the real-time joint prediction for the non-monotonic model is a very time-consuming operation, therefore we do not compute the AUC-ROC on the full test set but just on the random subset of 1000 logs. The results of these experiments are shown in Table 1. In the full log classification task, monotonic models demonstrates less impressive results than the non-monotonic one. That is an expected outcome since monotonic models satisfies the additional constraints and therefore have less expressive power. On the contrary, in the real-time classification task, the non-monotonic model can't make any reasonable predictions. The reason for such behavior is that the non-monotonic model learns to use a lot of 'benign' features while we forbid monotonic models to do so.

ACKNOWLEDGMENTS

Ekaterina Lobacheva has been supported by Russian Science Foundation grant 17-71-20072.

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

## A  EXPERIMENTAL SETUP

**Data format.** The set of possible event types in the logs contains 180 different elements. Each argument is represented as a set of tokens. For example, the filename `C:\Windows\374683.ini` corresponds to the set `['C', ':\', 'Windows' , '\', '374683', '.', 'ini']`. We use the vocabulary of 135 907 most popular tokens from the training data. As a result, each pattern is described with a vector of size 136 087 with counters for event types and tokens.

**Feature representation.** For pattern feature extraction we use a linear embedding layer with output of size 300. In addition, we use counter features for 382 of the most popular event groups in the training data. As a result, the feature representation for graph contains 682 elements.

**Classifiers.** For the baseline non-monotone model and deep monotone model we use a neural network with 4 hidden layers as a classifier. These layers have the following architecture (numbers of hidden units and nonlinearities): $600 \ tanh - 300 \ ELU - 100 \ ELU - 50 \ tanh$. We also try a one linear layer classifier and a min-max network with 10 MIN blocks, 20 neurons each.

**Loss function.** We use the stochastic version of the continuous $1 - AUC$ upper-bound as an objective in all experiments:

$$\mathcal{L}(B, M) = \frac{1}{|B| \cdot |M|} \sum_{b \in B} \sum_{m \in M} \max(0, s_b - s_m + 1)^2 \geq \frac{1}{|B| \cdot |M|} \sum_{b \in B} \sum_{m \in M} \mathbb{I}[s_b \geq s_m] = 1 - AUC$$

Where $B$ and $M$ are sets of benign and malicious items in a batch and $s_x$ is the suspiciousness predicted by the trained model on object $x$. Minimizing of this objective affords to obtain higher AUC-ROC value than minimizing of the the standard log-loss and could be computed with $O(|B| + |M|) \cdot \log(|B| + |M|)$ operations as the traditional AUC-ROC.

## B  INTERPRETATION OF PREDICTIONS OF THE MONOTONIC MODEL

Predictions of monotonic models are usually interpretable. In Figure 2 we demonstrate predictions of the deep monotonic model for three malicious logs alongside with the suspicious lines from these logs on which the model's prediction grows significantly.

The top picture corresponds to a Cryptocurrency Trojan Miner. Here the model notice the start of a specific service, writing to the auto-run and saving a specific miner URL and port number to the system register.

The middle picture represents a typical ransomware cryptor. Almost all the time of execution the program takes a new file from a filesystem, modifies it somehow and then changes the file extension by adding xoxoxo (that could be read as hohoho in a Cyrillic notation). An interesting observation is that the model increases the predicted suspiciousness while reading the whole sequence but the detection threshold is crossed after approximately 350 events. So, if we stop the process at this point, we will save the rest of a filesystem from encryption.

The bottom picture the inspected program has no malicious activity except for starting a legitimate powershell process with a very interesting parameter – a long base64 encoded string. This string is an obfuscated malicious script.

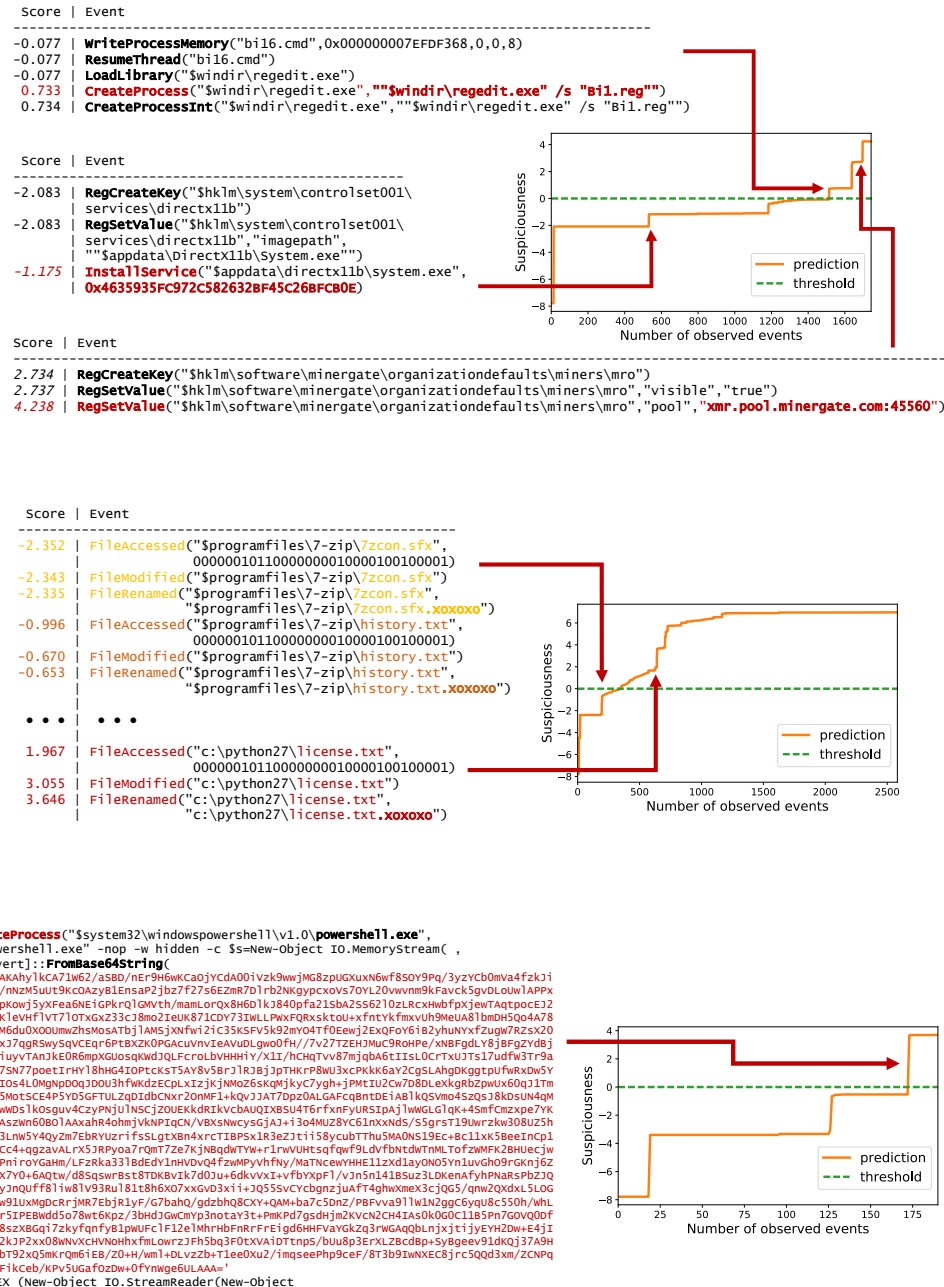

Figure 2: Predictions of the deep monotonic model for three malicious logs alongside with the suspicious lines from these logs.

