# OpenReview forum: "Monotonic models for real-time dynamic malware detection"
_ICLR.cc/2018/Workshop — Accept_

### Official Review · AnonReviewer2 · 2018-03-09

**Rating:** 7
**Confidence:** 3

**Review:**

The manuscript considers malware detection when following a piece of software run as online process. There is a good motivation: existing methods work in batch mode, and can be fooled compensating malign activity with lots of benign activities. They propose monotonic malware detection, where more execution can only increase how malign the software is. This feels somewhat faulty assumption as well since it could lead to false positives by software that often does something slightly malign. The introduction and motivation are excellent.

The method description suffers from brevity, but seems ok. In the expermients the monotonic approach works well on online setting as expected, while suffers a bit in batch mode.

Overall the presented idea is interesting and results are good. The abstract is clearly written.

---

> ### Author Response · Authors · 2018-04-20
> **Author's response**
>
>
> >This feels somewhat faulty assumption as well since it could lead to false positives by software that often does something slightly malign.
>
> Unfortunately, in malware detection, it is quite difficult to formally distinguish between the benign software that does something slightly malign and real malicious software. It is a common problem in a lot of other applications too. We agree that this assumption is not ideal in the real world, but from our experience and results, it is adequate.

---

### Official Review · AnonReviewer1 · 2018-03-11
**Interesting application of real-time malware detection**

**Rating:** 7
**Confidence:** 3

**Review:**

This paper proposes a method of real-time malware detection through a program's execution time. It assumes that the probability of being malicious increases monotonically as new lines of codes are added into the execution log. As a result, the risk of a benign program is expected to be stable, but that of a malicious program will increase until it passes a threshold. The paper applies the monotonic classifier (e.g., min-max network) and modifies the feature extraction process under the monotonicity constraint. Experiments show that in the full-log setting, the monotonic models are close to non-monotonic models. In the real-time setting, however, monotonic models perform much better than the non-monotonic models. The paper explains that "such behavior is that the non-monotonic model learns to use a lot of ‘benign’ features while we
forbid monotonic models to do so". But does that mean the experiments involve inconsistent feature sets? While non-monotonic model learns to use a lot of ‘benign’ features, but the real-time setting does not have such features. What if we totally forbid these features for both models? The paper should give more explains since the gap between the two models are big in the real-time setting. Moreover, the analysis in the appendix is very interesting. It will be even better if the paper can include the case when the monotonic model makes mistake. In the full-log setting, the monotonic model is worse than the non-monotonic model. Is it because it makes more false-positive predictions or vice-versa?

---

> ### Author Response · Authors · 2018-04-20
> **Author's response**
>
>
> >But does that mean the experiments involve inconsistent feature sets? While non-monotonic model learns to use a lot of ‘benign’ features, but the real-time setting does not have such features. What if we totally forbid these features for both models?
>
> Thank you for raising this point. We think the explanation of ‘benign’ features in the paper turned out slightly confusing and caused a misunderstanding.
>
> Here is what we mean when we say ‘benign’ feature.
> Consider some trained model which takes an execution log as an input and returns the predicted probability of maliciousness. For simplicity, assume that our log is a sequence of events without arguments and there are 100 possible types of events. In this case, we can represent each log with 100 features (counters for all event types) without information loss.  Here we do not care about the order of events because we do not use it in any of the models. A feature from this set may be ‘benign’ for the model: if we increase the value of this feature then the prediction will decrease. A subset of features can also be ‘benign’ for the model: if we increase values of all features from this subset then the prediction will decrease.
> The point is that this definition of ‘benign’ features depends on the model. In the non-monotonic model, we allow non-monotonic dependencies between features and predictions, therefore, it may have such ‘benign’ features. In the monotonic model, if the value of any feature increases the prediction can't decrease, therefore, ‘benign’ features are not possible in such model.
>
> Both non-monotonic and monotonic models use an execution log as an input in any scenario (real-time or full log). The only difference is that in the full-log scenario they use a full log and in the real-time scenario they use a prefix of a log (which is also a log but a shorter one). Therefore, a feature set is the same for both scenarios.
>
> >The paper should give more explains since the gap between the two models is big in the real-time setting.
>
> We were also very surprised by these results. From our experiments, we think that the main reason here is that the non-monotonic model is not restricted in its behavior in the middle of the log. Therefore it uses ‘benign’ features and possibly ‘benign’ groups of features which makes predictions unstable in time. We are working on more detailed experiments now.
>
> >Moreover, the analysis in the appendix is very interesting. It will be even better if the paper can include the case when the monotonic model makes mistake.
>
> Thank you for the suggestion. We will work on this.
>
> >In the full-log setting, the monotonic model is worse than the non-monotonic model. Is it because it makes more false-positive predictions or vice-versa?
>
> We also think that this question is important for understanding monotonic models. We will investigate it more but for now, the answer is both. We compared  ROC curves of the models, and the curve for the non-monotonic model is higher than curves for monotonic models everywhere. Also, we compared models with optimal thresholds, and the optimal non-monotonic model has higher TPR and lower FPR than optimal monotonic models.

---

### Official Review · AnonReviewer3 · 2018-03-12
**An interesting read**

**Rating:** 7
**Confidence:** 1

**Review:**

The paper discusses real time detection of malware using a monotonic network, for which the output probability is restricted to monotonically increase as more events become available. The paper is interesting to read, and presented well. I didn't fully understand why the non-monotonic network performs at a random level for real-time detection. The authors mention that this happens since non-monotonic network learn benign features. Some experimental result to elaborate this further would be helpful.

---

> ### Author Response · Authors · 2018-04-20
> **Author's response**
>
>
> >The authors mention that this happens since non-monotonic network learn benign features. Some experimental result to elaborate this further would be helpful.
>
> We also think that this point is very important and we are now working on such experiments.

---

### Author Response · Authors · 2018-04-20
**Update**

We would like to thank the reviewers for their time and effort to make our work better. To address the raised concerns, we updated the paper in the following ways:

1) We have explained the term ‘benign’ features more accurately.
2) We have fixed a typo in the loss function in appendix A.

We are also grateful for the suggested additional experiments. We are going to conduct them and present the results in the full version of the paper.

---

### Decision · Program_Chairs · 2018-03-20
**ICLR 2018 Workshop Acceptance Decision**

**Decision:**

Accept

**Comment:**

Congratulations, your paper was accepted to the ICLR workshop.